# Dark wing pigmentation as a mechanism for improved flight efficiency in the Larinae

Madeleine Goumas [1]✉

There are many hypotheses explaining the diversity of colours and patterns found in nature, but they are often difficult to examine empirically. Recent studies show the dark upperside of gliding birds' wings could reduce drag by decreasing the density of surrounding air. It may therefore be expected that species with darker wings have less efficient morphology than their paler counterparts. I conducted an analysis of the Larinae (gulls), which exhibit extreme variation in wing (mantle and wingtip) melanization, to test whether wing loading is a predictor of wing darkness. I found that, for each standard deviation increase in wing loading, mantle darkness is predicted to increase by 1.2 shades on the Kodak grey scale. Wing loading is also positively related to the proportion of black on wingtips. Furthermore, heavier species have lower aspect ratio wings, suggesting that dark wings have evolved to improve the trade-off between maneuverability and long distance flight.

[1] Centre for Ecology and Conservation University of Exeter, Penryn, UK. ✉email: m.goumas@exeter.ac.uk

A great number of hypotheses have been proposed to explain the diversity in patterns and colours in the animal kingdom, and biologists continue to debate the adaptive significance of pigmentation and markings. For example, the function of the black and white stripes of zebra has long been debated, with many popular hypotheses not withstanding analysis[1]. Some phenomena have received a lot of attention: it is well-established that camouflage plays an important role in disguising animals both from predator and prey[2]. Countershading, whereby the upperside of an organism is dark while the underside is pale, appears to allow animals to be concealed from both dorsal and ventral viewpoints: a pale underside blends in against a bright sky, while a dark upperside is less conspicuous against a backdrop of land or sea[3]. This colouration has evolved independently several times in various seabirds, suggesting an adaptive role[4].

While it is easy to understand the benefits of countershading, the function of other, similar, colour patterns is less clear. For instance, most species of the Larinae (gulls) are predominantly white, and the majority have a grey upperwing and back, which is often referred to as the mantle (in contrast to the typical definition, which refers solely to the part of the back below the nape)[5]. However, the nape, rump and tail are usually also white, making countershading an unsatisfactory explanation. Melanin, the pigment that produces grey, black or brown colours, is thought to be energetically costly to produce[6,7], so it is likely that this pattern has arisen and been maintained through selection. In addition, the wingtips of gulls commonly comprise a fully black region. Melanin has many functions, including protection of feathers against damaging ultraviolet radiation[8]. Indeed, Dufour et al.[9] recently found a positive effect of insolation (incident solar radiation) on Larinae mantle and wingtip pigmentation.

Wing colouration may, however, be explained by additional factors. Hassanalian et al.[10,11] showed that, because dark surfaces absorb heat, thus decreasing the density and increasing the viscosity of air flowing over them, the dark wing colour of many gliding birds may function to reduce drag by decreasing the amount of friction generated during flight. A recent analysis of seabirds[4] found that wing darkness is associated with morphology optimized for flight efficiency. However, this analysis was conducted at a broad scale, and morphology differs within and among taxonomic groups. Furthermore, for most seabird groups, wing loading and aspect ratio appear to be positively correlated, whereas previous estimates indicate that it is negatively correlated for Larids such as gulls[12]. This may be because of differences in foraging behaviour, with gulls generally taking shorter foraging trips than other seabirds[12]. High aspect ratio wings, possessed by birds that have a long wingspan compared to wing breadth, are associated with flight efficiency because they create less induced drag, whereas low aspect ratio wings permit better maneuverability[13]. Additionally, high wing loading, a bird's body mass per unit of wing area, results in a lower production of lift[13]. Birds must produce more lift than drag to remain in flight[14], and there is likely to be a trade-off between adaptation for maneuverability and for long distance gliding.

Gulls are an excellent system with which to test the hypothesis that wing darkness is related to flight efficiency. They share relatively similar morphology and ecology, but show striking variation in mantle darkness, the amount of black on wingtips and body size. For instance, one species, the Ivory Gull (*Pagophila eburnea*), is completely white, while others, notably large species such as the Great Black-backed Gull (*Larus marinus*) and Pacific Gull (*L. pacificus*), have almost black mantles and partially black wingtips. I hypothesized that the dark wing colouration of gulls functions to reduce drag, and that species with darker wings have evolved this pattern as a means of compensating for reduced flight efficiency derived from constraints on wing morphology.

Therefore, I predicted that the variation in wing darkness could be explained, at least in part, by species' wing loading, with relatively heavier species possessing darker mantles and wingtips.

I calculated the wing loading and aspect ratio of almost all recognized species of gull ($N = 50$; see Methods) and used published data on gull wing colour to investigate the effect of wing loading on wing darkness. I first considered mantle colour, which tends to be uniform across this body region. To ensure that any relationship between wing loading and mantle darkness was not simply a consequence of shared ancestry, I ran four different models, each assuming a different model of evolution, for comparison: a non-phylogenetic (NP) model (an ordinary least squares regression), a Brownian motion (BM) model (which assumes traits evolve via a random walk), an Ornstein-Uhlenbeck (OU) model (which constrains the evolution of traits towards an optimum) and Pagel's lambda model (which transforms the tree branches according to estimated phylogenetic signal in trait covariance). As the selection pressure generated by insolation[9] may obscure or confound an effect of wing loading, I incorporated this as a variable in the model. Insolation is directly related to the distance from the equator[15], thus I converted the centroid latitudes of each species' breeding and resident range to absolute values. I next tested whether the relationship between wing loading and wing pigmentation holds when considering the proportion of black on each species' wingtips. Finally, I assessed whether species with high wing loading have lower aspect ratio wings.

## Results

**Wing loading predicts an increase in mantle darkness**. Wing loading ranged from 0.002 to 0.007 g mm$^{-1}$ and was associated with an increase in mantle darkness across the four tested models. The OU model provided a significantly better fit of the data than either the NP or the BM model (Table 1). The lambda model did not fit the data significantly better than the NP model (Table 1). The phylogenetic half-life[16,17] estimated by the OU model was 0.026 million years, which suggests a small but non-trivial influence of phylogeny on the measured traits, and selection towards an optimal value of mantle darkness over evolutionary history (Fig. 1).

Both wing loading and absolute latitude had statistically significant effects on mantle darkness. The OU model estimated a positive increase on the Kodak Grey Scale (KGS) of $1.22 \pm 0.51$ SE for each standard deviation increase in wing loading (Table 1, Fig. 2a). Conversely, for each degree increase in distance from the equator, a reduction in mantle darkness equivalent to $-0.07 \pm 0.03$ SE on the KGS was estimated (Table 1, Fig. 2b).

**Wing loading is positively related to the proportion of black on the wingtips**. Consistent with the findings on mantle darkness, wing loading was associated with an increase in the proportion of black pigmentation on the wingtips (Table 2). For each standard deviation increase in wing loading, the ratio of black to non-black regions on the wingtips was predicted to increase by a factor of 1.4. To illustrate, at the equator (latitude = 0°) and at mean wing loading, the proportion of black on the wingtips was estimated by the model to be 0.84. An increase in wing loading of one standard deviation predicts a proportion of 0.88, and two standard deviations 0.91. Additionally, the amount of black on the wingtips is predicted to change by a factor of 0.95 for each degree increase from the equator. Mantle darkness and wingtip darkness have a correlation coefficient of 0.79.

**Heavier species have less efficient wing morphology for gliding**. Wing loading was negatively correlated with aspect ratio

**Table 1 The effect of wing loading and latitude on gull mantle darkness.**

| Model | Estimate ± SE (P-value) | Log Likelihood | LRT P-value |
|---|---|---|---|
| Ornstein-Uhlenbeck (OU) $\alpha = 27.13$ $\sigma^2 = 595.77$ | $\beta_0$: 10.38 ± 1.28 (<0.001) WL: 1.22 ± 0.51 (0.021) AL: −.07 ± 0.03 (0.011) | −125.15 | 0.013 (NP) <0.001 (BM) |
| Lambda $\lambda = 0.44$ (0 – 0.87) $\sigma^2 = 58.89$ | $\beta_0$: 10.16 ± 1.81 (<0.001) WL: 0.89 ± 0.52 (0.090) AL: −.08 ± 0.03 (0.003) | −127.09 | 0.134 (NP) <0.001 (BM) |
| Non-phylogenetic (NP) | $\beta_0$: 10.54 ± 1.18 (<0.001) WL: 1.34 ± 0.47 (0.006) AL: −.07 ± 0.02 (0.005) | −128.21 | — |
| Brownian motion (BM) $\sigma^2 = 275.45$ | $\beta_0$: 9.42 ± 3.93 (0.021) WL: 1.26 ± 0.51 (0.017) AL: −.06 ± 0.03 (0.017) | −133.57 | — |

Candidate models for selection are shown, with the results of likelihood ratio tests (LRT). The model with most support is listed first. The regression estimates are labelled as follows: $\beta_0$, intercept; WL, standardized wing loading; AL, absolute latitude (distance of the species' breeding and resident range from the equator). The following parameters for each phylogenetic model are also shown: $\sigma^2$, evolutionary rate (accumulated variance); $\alpha$, strength of selection toward the optimum (OU only); $\lambda$, covariance due to phylogeny (lambda only, with 95% confidence interval). N = 50 species.

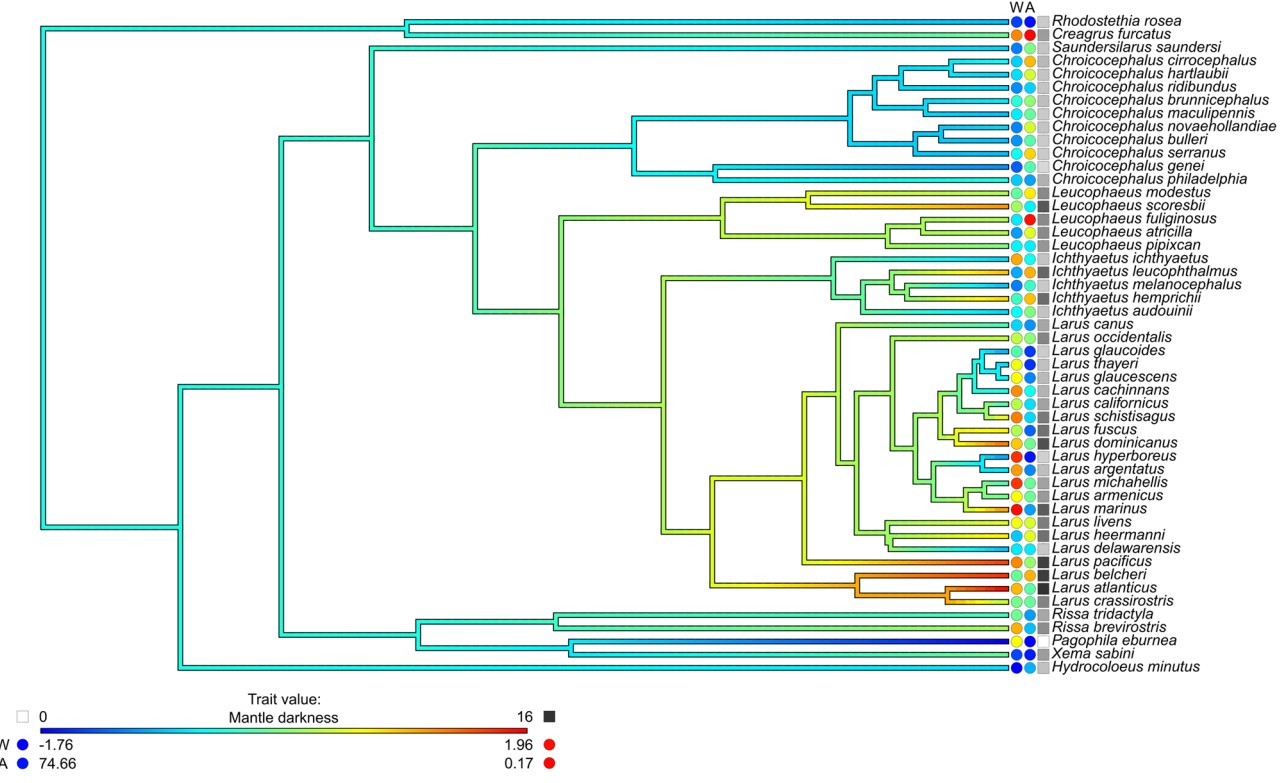

**Fig. 1 The evolution of mantle darkness in the Larinae and its relationship with wing loading and absolute latitude.** Branches show the estimated ancestral states of mantle darkness, with the current, observed states at the tips. The relative values of standardized wing loading (W) and absolute latitude (A, distance from the equator in degrees) are shown by coloured circles, along with the approximate mantle darkness of each species on the Kodak grey scale (squares). The height of the phylogenetic tree is 0.21 million years. Note: genus names of the original BirdTree phylogeny have been amended to reflect current nomenclature.

(estimate: $-0.39 \pm 0.12$ SE, $R^2 = 0.18$, $p = 0.002$; Fig. 3), which is likely to reduce flight efficiency during gliding flight[14].

## Discussion

Variation in colouration among closely related species is not often easily explained, and there can be many competing hypotheses. One such hypothesis was proposed by Hassanalian et al.[10], who found that a dark surface on the upper side of artificial wings reduces skin friction drag. Birds that spend a long time in flight, such as seabirds, could therefore be expected to benefit from dark wing pigmentation. Rogalla et al. corroborated this hypothesis and showed that darker birds' wings do indeed create less drag when exposed to radiation[4]. However, because the production of melanin, the pigment responsible for dark feather colouration, is likely to be costly[7,18], a trade-off can be expected. It is unlikely that species would produce dark feathers if the benefits they confer become superfluous. I proposed that species with a higher wing loading, which has a detrimental effect on flight efficiency, could be expected to have darker wings. Consistent with this expectation, wing darkness in the Larinae varies widely but tends to be darker in species with higher wing loading, regardless of ancestry.

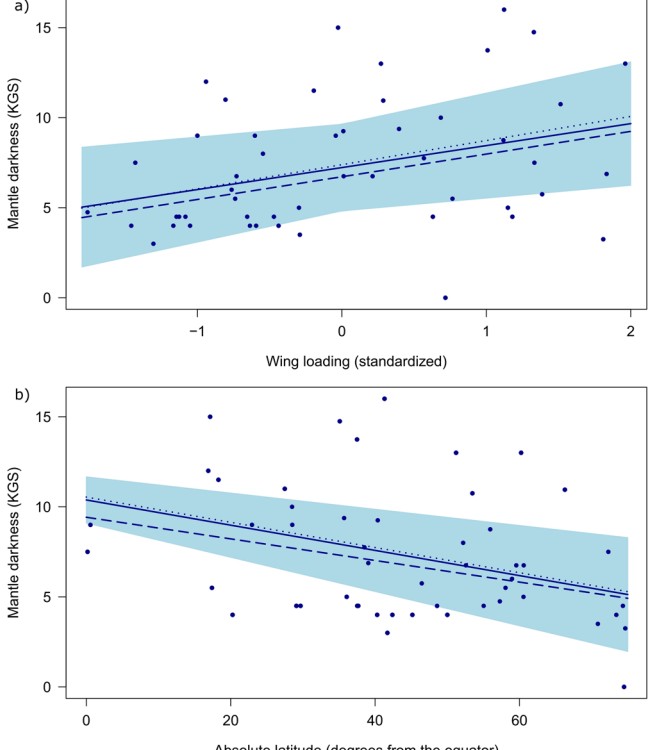

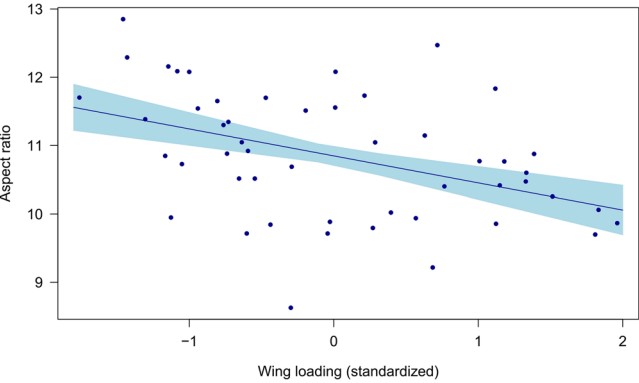

**Fig. 3 The relationship between wing loading and aspect ratio in the Larinae.** Species that are heavier relative to their wing area have a relatively broader wing chord. The shaded area depicts standard error. $N = 50$ species.

**Fig. 2 Model estimates of the effect of wing loading and latitude on mantle darkness, measured on the Kodak Grey Scale (KGS). a** Wing loading at 45 degrees from the equator; **b** absolute latitude (distance from the equator of species' breeding and resident ranges in degrees) at sample mean wing loading (0 SD). Predictions for Ornstein-Uhlenbeck (best-supported; solid line with standard error), non-phylogenetic (dotted line) and Brownian motion (dashed line) models of evolution are shown. $N = 50$ species.

**Table 2 The effect of wing loading on the proportion of black on the wingtips.**

|  | Estimate ± SE | Ratio | Z | P |
| --- | --- | --- | --- | --- |
| Intercept | 1.65 ± 0.44 | 5.20 | 3.73 | <0.001 |
| Wing loading | 0.34 ± 0.16 | 1.40 | 2.08 | 0.037 |
| Absolute latitude | −0.05 ± 0.01 | 0.95 | −5.24 | <0.001 |

Results of a beta regression with logit link function and phylogenetic covariance structure. The ratios signify, for the intercept, the ratio of black to non-black predicted by the model at the baseline values (mean wing loading and a latitude of 0°) and, for each predictor variable, the ratio of black to non-black predicted by one unit increase divided by the ratio at the baseline. $N = 49$ species.

Furthermore, gull species with a higher wing loading have lower aspect ratio wings. High wing loading among birds is principally driven by a faster increase in mass than wing area as body size increases, a pattern observed across the avian class (excepting hummingbirds)[19]. As wing area reliably increases with increasing body mass[19], a decrease in aspect ratio in gulls must occur as a result of wing breadth increasing faster than wing length. It therefore appears possible that darker mantles and wingtips have evolved in gull species that have become heavier as a means of compensating for the reduction in flight efficiency that results from possessing wings that are relatively short both in proportion to body mass and to wing breadth.

Although the finding that higher wing loading is associated with darker wings may appear to contradict the findings of

Rogalla et al.[4], who found that wing darkness is related to more efficient flight morphology, in actuality both studies support the central hypothesis, which is that dark wing pigmentation increases flight efficiency. In some seabird taxa, such as albatrosses, species may use multiple means of optimizing flight efficiency, by decreasing wing loading and increasing both aspect ratio and melanin pigmentation. This may be because these species spend longer in flight than do gulls, and benefit from minimizing drag rather than optimizing maneuverability. While high aspect ratio wings cause a reduction in induced drag during flight, they are not optimal for take-off[19]. It is likely that gulls, which spend longer on land and at rest than most seabirds[20], require wings optimized for rapid and regular take-off, which is a particular challenge for species with high wing loading.

Other factors besides those examined in this analysis will explain some of the variation in wing darkness observed across the Larinae. Dufour et al. showed that mantle darkness is also negatively correlated with the average temperature in the non-breeding range[9], suggesting that dark mantle colouration plays a thermoregulatory role, consistent with larger scale patterns in animal melanization[21]. The fact that relatively heavier (and thus larger) species, which can thermoregulate more effectively in cold temperatures, were more likely to have darker wings suggests that any effects of wing darkness on thermoregulation and flight efficiency are acting independently. It also seems unlikely that dark wing colouration evolved as a means of camouflage in gulls, since the larger species moult from cryptic brown juvenile plumage and have few predators as adults.

Because thermal effects on efficiency should be more apparent during unpowered flight, where flight speed is low and the wing surface is continuously exposed to solar radiation, Rogalla et al. predicted that flight mode (i.e. flapping vs. gliding) would have influenced the evolution of dark wing pigmentation[4]. However, they did not find support for this prediction. All gull species were listed as employing flapping flight, despite gliding being a common flight strategy in this clade[22]. It may still be possible that flight mode has an effect on wing darkness, as the degree to which gliding flight is used varies among species, with larger species tending to glide more frequently than smaller ones[22,23]. Nevertheless, it is plausible that melanin could also increase efficiency during flapping flight[4,24].

The average length of foraging trips undertaken by each species could potentially influence wing darkness, as species that remain airborne for longer will have a greater need to optimize long distance flight rather than maneuverability during take-off. Migration distance may also be a key factor. For example, Herring

Gull (*L. argentatus*) and Lesser Black-backed Gull (*L. fuscus*) have similar wing loading and breed at similar latitudes, but the latter has a far darker mantle. Lesser black-backed gulls migrate further, spending the winter near the equator[5]. Migration distance and absolute latitude are, however, likely to be correlated: species breeding nearer the poles are more likely to be migratory and have longer migration distances[5]. Lesser black-backed gulls and species with similar migration ecology may benefit from having darker plumage to protect their flight feathers from solar radiation, or from wind abrasion[25], or perhaps because long distance migrants accrue greater benefits from reducing drag. These hypotheses are not mutually exclusive. Additionally, it is plausible that species nearer the equator gain more utility from dark pigmentation reducing drag owing to the greater insolation in these regions[15]. A method that can disentangle the effects of the potential protective and thermal benefits of melanism would greatly add to our understanding of the evolution of dark wing colouration in gulls and other birds.

Differences in colouration between closely related species could also act as a means of species recognition, preventing or reducing potentially costly hybridization, or as a result of sexual selection for honest signals of mate condition[26]. Hybridization is relatively common in gulls, yet there remain distinct phenotypes[27]. Moreover, gulls are sexually monochromatic, indicating that the two sexes do not experience differential selection on plumage colouration. Experimental studies would need to be conducted to assess whether plumage colour is an important marker of species identity and condition in gulls. The present study suggests that, at least in part, variation in colour has evolved for reasons other than selection pressures imposed by conspecific or heterospecific visual systems, as is the case in mate recognition and camouflage.

These results highlight the conditions under which colouration can function to increase fitness in ways that are unrelated to visual ecology. It is possible that colour may function in a similar, as yet undiscovered, way in other animal taxa. Furthermore, there may be potential for flight efficiency of aircraft to be improved by imitating the dark feather pigmentation that is widespread in seabirds. Full validation of the hypothesis that wing darkness is related to flight efficiency will require measuring the energetic costs incurred by living birds[24].

## Methods

**Mantle and wingtip darkness**. I used the mantle and wingtip darkness values provided in the supplementary material of Dufour et al.[9] in their analysis of the relationship between gull wing colouration and climactic conditions. They follow the Kodak grey scale (KGS) method of measuring gull mantle darkness, which has 20 units from 0 (white) to 19 (black), and calculated the mean values provided by Olsen[5] and Howell and Dunn[28]. The mantle in the context of gulls refers to the back, scapulars, secondary feathers, secondary coverts and tertials. Wingtip darkness is calculated as the proportion of black on the upperside of each species' hand region (primary feathers, primary coverts and alula) and was measured from colour photographs[9]. I only considered full species (N = 51) according to Jetz et al.'s BirdTree[29] in my analysis so, where wing darkness measurements were provided for subspecies, I calculated the mean for the species. American Herring Gull (*L. smithsonianus* or *L. argentatus smithsonianus*) is treated as conspecific with European Herring Gull (*L. argentatus*). There was no wingtip measurement available for Glaucous-winged Gull, since this species has variable grey wingtips (i.e. no black component) and was excluded from Dufour et al.'s analysis of wingtips[9].

**Wing loading**. Wing loading requires an estimate of wing area, which is not widely available in the literature. I therefore used the information on hand length (H, known as wing length) and first secondary feather length (S) provided by the AVONET database[30] and calculated wing area with the following equation:

$$Wing\ area = (Wingspan - 2 \cdot H)^*S + 2(H \cdot S \cdot 0.5)$$

This method has been used in a study of common sandpipers (*Actitis hypoleucos*)[31] and, while it underestimated wing area compared to planforms (drawings of the outline of a bird with the wings extended), the two measurements were significantly and highly correlated (R = 0.83, p < 0.01). I obtained wingspan measurements from Olson[5]. As only ranges were provided, I took the central value. These were highly correlated with both the minimum and maximum wingspan

(both R > 0.98). There was no wingspan measurement available for Relict Gull (*Ichthyaetus relictus*), and therefore this species was not included in the analysis. The final sample size was therefore 50, apart from the wingtip analysis where the sample size was 49 due to the exclusion of Glaucous-winged Gull (see above)[9]. I divided the wing area calculation by the body mass (also from AVONET[30]) to obtain the final estimate of wing loading. To aid interpretation of the regression coefficient, I standardized this variable such that the mean takes a value of 0.

**Covariate: absolute latitude**. AVONET[30] also provides the centroid latitudes (the geometric centre) of the breeding and resident ranges for each species. I converted these to absolute values, termed "absolute latitude", to obtain a variable to include as a covariate in the analysis. This variable denotes the distance from the equator in degrees and is a proxy for the amount of insolation experienced by each species[15].

**Statistics and reproducibility**. To control for shared ancestry, I downloaded the maximum clade credibility tree of Lari from the supplementary material of Jetz et al. (2012) and pruned the tree to include only the Larinae. I used the package phylolm[32] in R v. 4.1.2[33] to run phylogenetic generalized least squares regressions of the relationship between wing loading and mantle darkness with absolute latitude as a covariate. As there is some contention in the literature about which type of phylogenetic model should be used[34,35], I ran four different models for comparison (see Introduction). I compared the fit of the two models with fewer restrictions (NP and BM) with the two models with an extra parameter (OU and lambda) and selected the best supported model for inference while also reporting the results of the other models. In addition, to assess whether selection of the OU model over the BM model was not simply a result of bias in the tree, I simulated traits evolving along the tree through both a BM and an OU process 500 times and compared the difference in AIC in these simulations with the difference in AIC between the OU and BM model.

The phylogenetic tree of the Larinae, with ancestral values of mantle darkness estimated along the branches, was produced using contMap in the package phytools[36].

As many gull species have discrete black regions on their wingtips, wingtip darkness was measured by Dufour et al. as the proportion of black on this section of the wing[9]. Proportion data is best modelled as a beta distribution[37]. There are few software packages that allow the analysis of phylogenetic data with a beta distribution, and I therefore used glmmTMB, which has an extension allowing incorporation of a phylogenetic covariance matrix for comparative analysis[38].

In beta regression, values must be greater than 0 and less than 1 and it is common practice to transform extreme values to lie within this range (there were too few extreme values to require a zero-one inflated beta regression)[39]. Species whose wingtips lack a black component (N = 5) were assigned the value $1 \times 10^{-4}$ and those with fully black wingtips (N = 1) were assigned $9.999 \times 10^{-1}$. Standardized wing loading and absolute latitude were entered as predictor variables. Model fit was assessed using the R package DHARMa, to check that there was no deviation of residuals from the expected distribution.

To aid interpretation of the logit scale used by beta regression, I exponentiated the coefficients provided by the model output, which gives ratios; these are analogous to odds ratios in logistic regression. The intercept value is the ratio of black to non-black regions on the wingtip for a species predicted at the equator and with mean wing loading. The values for each predictor variable are the amount required to yield the ratio of black to non-black on the wingtips for each unit increase in x as predicted by the model. These ratios can then be converted back to proportions.

**Aspect ratio**. The aspect ratio of each species was calculated by dividing the square of the wingspan by the wing area. I conducted a linear regression with aspect ratio as the dependent variable and standardized wing loading as the independent variable.

**Reporting summary**. Further information on research design is available in the Nature Research Reporting Summary linked to this article.

## Data availability
The data used in this analysis can be found at https://doi.org/10.5281/zenodo.7156454[40].

## Code availability
The code used to run these analyses can be found at https://doi.org/10.5281/zenodo.7156454[40].

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

## Author contributions

M.G. conceived the idea, analyzed the data and wrote the manuscript.

## Competing interests

The author declares no competing interests.
