## [Peer Review File · Communications Biology]

Reviewers' comments:

Reviewer #1 (Remarks to the Author):

The authors use comparative analyses to test the hypothesis that dark wing coloration aids flight in gulls, and find the predicted correlation between wing loading and dark wing (mantle) color, suggesting that dark wing color may aid heavier species in flight. This is an interesting and timely hypothesis, the analyses are well done, and the paper is clearly written. However, the data on wing color is incomplete- specifically, the data from Dufour et al. that they use for wing color is on "mantle"* color, and only includes about half of the wing (see fig. 1 in Dufour et al) and excludes the outer wing area, including the wingtips. This is a significant shortcoming, as the wingtips produce the greatest turbulence during flight and thus strongly contribute to drag. Furthermore, Rogalla et al. (2019, J.Roy. Soc. Interface 16:20190032) showed that black wingtips on a light-coloured wing produced significant temperature differentials that enhanced lift. The authors here thus need to examine more than this single variable on mantle color. One idea would be to assign an overall wing color based on an average of mantle color and percentage black wingtip, and/or analyze them separately. Until this is done, these analyses are incomplete.

Title: Not clear what tradeoff is- how do wing morphology and color trade off? Seems like birds compensate for weight with color, but they don't directly trade off with one another.

L60: Should also mention how you collected data on wing darkness

*as a sidenote, the authors should clarify in the methods that they are using an unusual definition of the word "mantle", which typically only refers to the back of the neck.

Reviewer #2 (Remarks to the Author):

Dear Editor,

The study reports a comparative association between wing loading and wing colour in gulls. As wing loading is tightly linked to body size, this also means there is an association between body size and wing colour. This is opposite to a comparative association demonstrated in previous work on seabirds, where darker wings have more efficient flight morphologies.

The author interprets this as being in support of the colour-based drag reduction hypothesis, suggesting that dark colour 'compensates' for having less efficient wings. I have no issue with this argument - there is no reason why correlations between phenotypes should be consistent across all avian groups. However, no evidence is presented to support this interpretation over the many other possible explanations for why wing colour might be related to wing loading/body size. The author mentions some of these in the discussion, including how they relate to the other association observed between wing colour and range latitude, but there is no attempt to empirically discriminate between the potential explanations of wing darkness, including camouflage, social or sexual signalling, UV protection and thermoregulation. Plausible arguments for why all these functional hypotheses might predict an association between wing loading and darkness could be made.

Thus, I think that while the methods are sound and appropriate, the result is a bit light for a general biology journal. This could be rectified by developing a framework of comparative predictions and collecting and analysing additional data on variables such as UV load (global databases of UV irradiance are available), frequency of flight modes, foraging behaviour, social behaviour, longevity, wing colour pattern, seasonal weather in winter and breeding ranges, and perhaps other key traits for which data is available.

I think the evidence emerging for the link between wing colour and flight is fascinating and demands full investigation. The gulls seem like a great group to investigate. I hope the author decides to expand the study, but if not then I think it still is a valuable result that should be published, just perhaps in a more specialist journal.

Yours sincerely,

Will Allen

L78 add reference to Hansen 1997. Should the half life be in units of time?

L101 I find it confusing to have mantle darkness plotted as both branch colour and a tip icon (especially with different colour scales)

L105 Really extant Larinae have diversified only in the last 200,000 years?!

L209 I think it reasonable to acknowledge and briefly justify taking colour measurements from drawings in field guides.

L213 Did you remove the taxonomically imputed species from the phylogeny? If not this should be acknowledged.

Response to reviewers' comments

Please see replies in italics to all comments by the reviewers. All changes to the manuscript are displayed with 'track changes' (there is also a 'clean' version).

Reviewer #1 (Remarks to the Author):

The authors use comparative analyses to test the hypothesis that dark wing coloration aids flight in gulls, and find the predicted correlation between wing loading and dark wing (mantle) color, suggesting that dark wing color may aid heavier species in flight. This is an interesting and timely hypothesis, the analyses are well done, and the paper is clearly written.

Thank you for your time reviewing the manuscript and for your kind comments.

However, the data on wing color is incomplete- specifically, the data from Dufour et al. that they use for wing color is on "mantle"* color, and only includes about half of the wing (see fig. 1 in Dufour et al) and excludes the outer wing area, including the wingtips. This is a significant shortcoming, as the wingtips produce the greatest turbulence during flight and thus strongly contribute to drag. Furthermore, Rogalla et al. (2019, J.Roy. Soc. Interface 16:20190032) showed that black wingtips on a light-coloured wing produced significant temperature differentials that enhanced lift. The authors here thus need to examine more than this single variable on mantle color. One idea would be to assign an overall wing color based on an average of mantle color and percentage black wingtip, and/or analyze them separately. Until this is done, these analyses are incomplete.

The original purpose of the paper was to attempt to explain the diversity in "mantle" colouration. However, I do agree that the phrasing of "wing darkness" was (unintentionally) misleading and that the conclusions cannot be as robust without considering the wingtip area. I have therefore now conducted an analysis of this section of the wing.

Title: Not clear what tradeoff is- how do wing morphology and color trade off? Seems like birds compensate for weight with color, but they don't directly trade off with one another.

I have changed the title.

L60: Should also mention how you collected data on wing darkness

I have now mentioned here that I used already-published data.

*as a sidenote, the authors should clarify in the methods that they are using an unusual definition of the word "mantle", which typically only refers to the back of the neck.

This has now been clarified at the first mention of "mantle" (second paragraph of the introduction). The exact parts of the birds' anatomy the term refers to is also now explained in the methods.

Reviewer #2 (Remarks to the Author):

Dear Editor,

The study reports a comparative association between wing loading and wing colour in gulls. As wing loading is tightly linked to body size, this also means there is an association between body size and wing colour. This is opposite to a comparative association demonstrated in previous work on seabirds, where darker wings have more efficient flight morphologies.

The author interprets this as being in support of the colour-based drag reduction hypothesis, suggesting that dark colour 'compensates' for having less efficient wings. I have no issue with this argument - there is no reason why correlations between phenotypes should be consistent across all avian groups. However, no evidence is presented to support this interpretation over the many other possible explanations for why wing colour might be related to wing loading/body size. The author mentions some of these in the discussion, including how they relate to the other association observed between wing colour and range latitude, but there is no attempt to empirically discriminate between the potential explanations of wing darkness, including camouflage, social or sexual signalling, UV protection and thermoregulation. Plausible arguments for why all these functional hypotheses might predict an association between wing loading and darkness could be made.

Thus, I think that while the methods are sound and appropriate, the result is a bit light for a general biology journal. This could be rectified by developing a framework of comparative predictions and collecting and analysing additional data on variables such as UV load (global databases of UV irradiance are available), frequency of flight modes, foraging behaviour, social behaviour, longevity, wing colour pattern, seasonal weather in winter and breeding ranges, and perhaps other key traits for which data is available.

I think the evidence emerging for the link between wing colour and flight is fascinating and demands full investigation. The gulls seem like a great group to investigate. I hope the author decides to expand the study, but if not then I think it still is a valuable result that should be published, just perhaps in a more specialist journal.

Yours sincerely,

Will Allen

Thank your thorough review of the manuscript and for your thoughtful comments, which are much appreciated. Unfortunately, I do not have the means to expand the study further, but I hope it will generate further conversation and research on this interesting topic. I have added some greater context to the discussion.

L78 add reference to Hansen 1997. Should the half life be in units of time?

I have now added this reference, as well as a more recent one that uses the term "half-life" rather than the original phrasing "half-time". Units have also been added.

L101 I find it confusing to have mantle darkness plotted as both branch colour and a tip icon (especially with different colour scales)

I have amended the figure such that the mantle darkness is plotted after the tip icons as squares to make it clearer (I thought it was informative/interesting to show the "actual" colours (plotting the branches as black and white did not show up the transitions well)).

L105 Really extant Larinae have diversified only in the last 200,000 years?!

I was also surprised by this, but, yes, it is true! I have double-checked with a different package (phytools as opposed to phylolm) to be sure.

L209 I think it reasonable to acknowledge and briefly justify taking colour measurements from drawings in field guides.

*Only mantle colour measurements for one species of the 50 (*Larus novaehollandiae*) were taken in this way. As such, and as this is explained and justified in the paper cited, I do not believe it is necessary to go into details in the present study.*

L213 Did you remove the taxonomically imputed species from the phylogeny? If not this should be acknowledged.

*I have not altered the phylogenetic tree in any way. I have amended this sentence, and hope it is clearer now. If this is referring to *Larus novaehollandiae* (as mentioned above), removing this species makes no difference to the results: at the published decimal places, the estimates and p-values are the same.*

REVIEWERS' COMMENTS:

Reviewer #1 (Remarks to the Author):

The authors have responded well to the comments of myself and the other reviewer. However, the available evidence (Hassanalian, Rogalla, etc.) indicates that dark wing color is beneficial for gliding flight, and whether it also benefits other flight modes has not yet been determined. Gulls mostly use flapping flight (as far as I know), so the authors should mention this current lack of knowledge, and temper their claims accordingly.

Response to reviewer comments – COMMS BIO 22-1525A

Reviewer #1 (Remarks to the Author):

The authors have responded well to the comments of myself and the other reviewer. However, the available evidence (Hassanalian, Rogalla, etc.) indicates that dark wing color is beneficial for gliding flight, and whether it also benefits other flight modes has not yet been determined. Gulls mostly use flapping flight (as far as I know), so the authors should mention this current lack of knowledge, and temper their claims accordingly.

MG response:

Gulls do use gliding flight, and there are several publications on their use of gliding flight (I have cited just one of these for succinctness, choosing the one I feel is most relevant here, but one of the studies reports herring gulls gliding continuously for over 20 minutes). I was surprised when I saw that Rogalla et al. (2021) categorised gulls as employing flapping flight rather than flap-gliding or glide-flapping. They did not find an effect of flight mode (i.e. flapping vs gliding) in their analysis of its effect on wing colouration.

I have added a paragraph to the discussion to explain and discuss this in more detail.